# Response of Apricot Fruit Quality to Protective Netting

Pablo Melgarejo [1], Pilar Legua [1,*], Rafael Martínez-Font [1], Juan José Martínez-Nicolás [1], Joaquín Sánchez Soriano [2], Ángel A. Carbonell-Barrachina [3] and Francisca Hernández [1]

1. Plant Science and Microbiology Department, High Polythechnic School of Orihuela (EPSO) Miguel Hernandez University, Ctra. Beniel Km 3.2, 03312 Orihuela, Spain; pablo.melgarejo@umh.es (P.M.); rafa.font@umh.es (R.M.-F.); juanjose.martinez@umh.es (J.J.M.-N.); francisca.hernandez@umh.es (F.H.)
2. Statistics, Mathematics and informatics Department, High Polythechnic School of Orihuela (EPSO) Miguel Hernandez University, Ctra. Beniel Km 3.2, 03312 Orihuela, Spain; joaquin@umh.es
3. Agro-Food Technology Department, High Polythechnic School of Orihuela (EPSO) Miguel Hernandez University, Ctra. Beniel Km 3.2, 03312 Orihuela, Spain; angel.carbonell@umh.es
* Correspondence: p.legua@umh.es; Tel.: +34-96674-9669

**Abstract:** The cultivation of fruit trees in protected environments is a technique that has been developed in recent years for peaches and nectarines, but not for apricots. This study was conducted to investigate the chemical composition of the fruits and their quality indices variations of the variety 'Mikado' as a function of its cultivation under a protective net or outdoors. As a practical agronomic assay, a homogeneous experimental plot was used in this study, where half of the apricot trees were cultivated under protective netting, and the other half without it. The data showed that for the total yield, no statistically significant differences were found with respect to 'Mikado' trees cultivated with or without a net. The trees cultivated under protective nets showed higher fruit weights. The results showed that the technique of using protective nets for the cultivation of extra-early apricot 'Mikado' is a profitable alternative for farmers, and their use does not affect fruit production or quality.

**Keywords:** antioxidant activity; apricots; phenolic compounds; net; sugars

## 1. Introduction

Apricots (*Prunus armeniaca* L.) are cultivated in temperate climates around the world. The main producing countries, in order of importance, are Turkey, Uzbekistan, Iran, Algeria, Italy, Pakistan, Spain, France, Egypt, and Ukraine. All of these countries are responsible for 72% of the world production, established at around 3,881,204 t [1] with a large number of local varieties [2]. Fideghelli and Della Strada [3] reported that from 1980 to 2007, 563 new apricot cultivars plus 61 hybrids (apricot × plum, plum × apricot) had been listed.

In Spain, the main apricot-producing area is located in the Region of Murcia (Southeastern Spain). In recent times, the efforts made in this region to investigate the genetic improvement of the species has been evidenced, especially associated to obtaining varieties that are resistant to Sharka disease (Plum Pox Virus-PPV), although research has also been conducted for decreasing the needs of cold conditions and for increasing productivity and precocity, among other physical characteristics, which have aimed to satisfy the needs of the fresh-consumption market and industry [4–7]. In this context, the region of Murcia has demonstrated an important varietal reconversion in the last 10 years, where the traditional apricot varieties have been relegated to very limited areas, while the new varieties occupy most cultivation areas, both traditional and new locations [7]. The new cultivation areas for apricot trees are located in warm coastal areas, thanks to the low winter cold needs of the new apricot varieties, which have allowed for a 15-day crop advance over traditional areas. This varietal change and the expansion of the crop to coastal areas, has allowed the first apricot fruits to be harvested in mid-April, which is a new and promising window for the early commercialization of this fruit, as it is possible to set very high prices at this time.

The cultivation of fruit trees in protected environments is a technique that has been developed in recent years for different fruits such as peaches and nectarines, but not for apricots [8]. In apricot trees, previous practical assays showed that greenhouse cultivation techniques had low production problems due to the limited bloom of the trees (unpublished data). Thus, for extra-early apricots varieties such as 'Mikado', which can attain a high economic value in the market, it is of great interest to study its fruit behavior under protected growing conditions, as this environment can protect the apricot fruit from adverse weather such as frost, winds, and/or hail, a frequent event in the region of Murcia, at times close to the apricot harvest season.

In this context, Melgarejo et al. [9] indicated, as the main advantages of protective net cultivation, the increase in plant growth during the first years; the minimization of hail, frost, and wind damage, which in turn may reduce the insurance costs paid, thus reducing the cost of cultivation; and the reduction of pests. In addition, Trénor et al. [10] and Martínez [11] indicated that the shade produced by the nets allows obtaining a more suitable environment for plants and fruits by reducing water and soil losses due to evaporation and wind drag, respectively. Nevertheless, despite these advantages, the decision to cultivate apricot or not under protective nets is complex, even with the currently available information. The use of nets could also be disadvantageous, due to a higher initial cost of the plantation and, above all, the possibility of negatively affecting the floral differentiation and the development, quality and color of the fruit. In this context, although some researchers have studied apricot quality, antioxidants, and polyphenols compounds [6,12,13], to our knowledge, there are no reports that have focused on the influence of protective net cultivation on these apricot fruit compounds.

Thus, this work aimed to investigate the chemical composition and the quality indices of apricot fruits of the variety 'Mikado' as a function of their cultivation under a protective net or outdoors. This work can be considered as the second phase of an initial practical agronomic study, which demonstrated the viability of the cultivation of apricots under a net without negative impacts on their development and flowering (unpublished data). Thus, once the viability of apricot cultivation under a protective net was demonstrated (fruit production), the fruit quality of the apricots obtained inside and outside the net were analyzed and compared. For the quality of the fruit, the physical parameters (color, firmness, fruit weight and seed, etc.), acidity, total soluble solids, antioxidant activity, organic acids, sugars, and total phenols contents, were analyzed.

## 2. Materials and Methods

### 2.1. Apricot Plants and Experimental Conditions

The plant material used in this work was an extra-early cultivar of *Prunus armeniaca*, namely 'Mikado' grafted onto Mirabolan 29/C (*Prunus cerasifera* (L.) C.K. Schneid.). The cultivar 'Mikado' COV CEE N°2014/2511 was obtained directly from a private breeder (PSB Producción Vegetal S.L., Murcia, Spain). The 4-year old apricots trees were placed in a commercial farm located in Ojós (120 m altitude) located in the Region of Murcia (Spain).

The protective net used in this study was fixed onto a galvanized steel tube structure (30 × 40 m). This structure was covered by a net on top and on the sides. The net used was a white high-density polyethylene mosquito net, with 6 × 6 and 6 × 9 strands cm$^{-1}$ for the ceiling and sides, respectively. The main properties of the mosquito net used were: a 20% reduction in wind, 95% light transmission, and 10% interior shading. The side nets could be rolled up to 3 m from ground level. The apricot trees were placed in eight rows composed by 10 trees each with 5 × 3 m separation between trees.

To maintain homogeneous conditions, in the same experimental plot, an additional eight rows of apricot trees without nets were selected. The eight rows without nets were used as control trees for comparing and studying the influence of the net on apricot quality. The control trees were set-up under the same parameters as netting trees (10 trees per row (5 × 3 m)). For both apricot tree samples, with or without the net, a randomized experimental design was used with four replication lots each. Each lot was composed

of five trees located in the center rows to avoid the border effect. Apricot fruits from each cropping system (without net and under the net) were manually harvested at the end of April, when commercial maturity was reached. The fruits were transported in ventilated plastic boxes and conserved at constant conditions (5 °C and 90% relative humidity) until used. The experimental assays were carried out less than three days after harvest. The number of fruits used for each assay is described below in the specific methodological section.

The farm soil had a clay-loam texture. The cultivation conditions were homogeneous for all the apricot trees both inside and outside the net. The irrigation system utilized was drip irrigation, with 3 self-compensating drippers and anti-drainage of 4 L/h per tree. All the trees were healthy. Climatic conditions and crop management were homogeneous for all apricot trees.

### 2.2. Measurements

### 2.2.1. Plant Yield and Physical Variables of the Fruit

For the total yield (kg tree$^{-1}$) the fruit boxes were weighed and the number of fruits counted. Later, 15 apricot fruits per experimental lot (4 lot × 2 crop system) were selected to determine the physical variables, which was measured according to Melgarejo et al. [4]: fruit and stone weights, fruit size, pulp thickness, flesh firmness, external color, and pulp yield. A total of 120 apricot fruits were used, with 60 fruits from each crop system.

### 2.2.2. Maturity Index (MI), Acidity, and Total Soluble Solids (TSS)

Once the yield and the physical variables of the fruits were determined, the apricot samples were divided into two subsamples composed of 30 fruits each. One subsample was used to determine the MI, total acidity, and TSS content, according to the methodology described by Melgarejo et al. [4]. While the other subsample was immediately frozen in liquid nitrogen, to be subsequently free-dried (Alpha 2–4; Christ, Osterode am Harz, Germany) for 24 h at 0.220 mbar. Once freeze-dried, the samples were crushed into powder and packed under vacuum to determine the organic acids, sugars content, antioxidant activity, and phenols following the Madrau et al. [14] methodology. The experimental results are shown as mean values ± standard deviation.

### 2.2.3. Sugars and Organic Acids Profile

Freeze-dried apricot fruit samples (0.5 g) were added to 10 mL of phosphate buffer 50 mM, pH = 7.8 helped with a polytron homogenizer (IKA Labotech- nik, Staufen, Germany). The resulting mixture was centrifuged for 20 min at 15,000 rpm, at 4 °C (Sigma 3–18 K; Sigma, Osterode am Harz, Germany). After that, 1 mL of supernatant was filtered through a 0.45 μm Millipore filter and 10 μL aliquot of the supernatant was used to quantify organic acids and sugars following the methodology described by Melgarejo et al. [4]. A column and a pre-column (Supelcogel TM C-610H column 30 cm × 7.8 mm and Supelguard 5 cm × 4.6 mm; Supelco, Bellefonte, PA, USA, respectively) were used, along with the standards of sugars (sucrose, glucose, and fructose) and organic acids (citric and malic), which were obtained from Sigma (St Louis, MO, USA). Analyses were carried out in triplicate and the results were expressed as g kg$^{-1}$ fruit weight (fw).

### 2.2.4. Antioxidant Activity (ABTS$^+$, DPPH$^\bullet$ and FRAP Methods) and Total Polyphenols

The solvent used for the antioxidant activity analysis was prepared as described by Wojdyło et al. [9]. Three methods were used for determine the free radical activity: ABTS$^+$, DPPH$^\bullet$, and FRAP, based on specific bibliography [15–17]. The antioxidant activity was determined using a UV–Vis spectrophotometer (Helios Gamma model, UVG 1002E) by measuring the absorbance variations at 734 nm after 6 min for ABTS$^+$, at 515 nm after 10 min for DPPH$^\bullet$, and at 593 nm after 10 min for FRAP. Analyses were run in triplicate, and the results were expressed as mmol Trolox kg$^{-1}$ fw.

Total polyphenols (TP) were measured according to described by Singleton et al. [18] by means of the Folin–Ciocalteu reagent. The absorption measurement was carried out at 765 nm by a UV-Vis Uvikon XS spectrophotometer (Bio-Tek Instruments, Saint Quentin Yvelines, France), previously calibrated with a concentration range between 0 and 0.25 g gallic acid equivalents (GAE) $L^{-1}$. Analyses were run in triplicate and results were expressed as mg of GAE 100 $g^{-1}$ fw.

### 2.2.5. Sensory Evaluation with Trained Panel

A trained panel was used to describe the fresh apricot fruits for its sensory evaluation. The methodology adopted was according to Melgarejo et al. [4]. In total, eight trained panelists (4 females and 4 males) aged 20–55 years old participated. Diverse questionnaires were used to conduct the sensory evaluation. In each questionnaire, panelists were asked to evaluate the intensity of the fruit attributes. Panelists used an 11-point scale for the evaluation, with 0 = extremely low intensity, 5 = regular intensity, and 10 = extremely high intensity.

### 2.3. Statistical Analyses

The software package SPSS 25.0 for Windows (SPSS Inc., Chicago, IL, USA) was used to carry out the statistical analyses. The mean comparisons were analyzed by analysis of variance (ANOVA). The Fisher's least significant difference (LSD) procedure at 95.0% confidence level was also applied. When the variables analyzed meet the hypotheses of normality and homogeneity of variances, the comparison of means of the two treatments was analyzed using the Student's t test. When the variables analyzed show non-parametric behavior, the non-parametric Mann–Whitney U test was used for two independent samples.

## 3. Results

The experimental results (Table 1) showed that the total yield (kg $tree^{-1}$), size fruit, average weight fruit, firmness, and thickness of the pulp, seed weight, yield in the pulp, TSS, acidity, and MI of apricot 'Mikado' were not influenced by the cultivation under a protective net.

**Table 1.** Physical and chemical properties of 'Mikado' apricot cultivar grown under a protective net and without a net. Mean values $\pm$ standard deviation ($n$ = 30).

| Variable | 'Mikado' under a Protective Net | 'Mikado' without a Net | ANOVA |
|---|---|---|---|
| Total yield (kg $tree^{-1}$) | 31.64 ± 1.75 | 29.80 ± 3.06 | ns |
| Fruit weight (g) | 61.24 ± 7.63 | 60.93 ± 8.02 | ns |
| Ø equatorial (mm) | 45.32 ± 1.98 | 46.22 ± 2,23 | ns |
| Fruit height (mm) | 47.48 ± 2.33 | 47.70± 2.66 | ns |
| Firmness (kg $cm^{-2}$) | 2.07 ± 1.30 | 1.72 ± 1.09 | ns |
| Pulp thickness (mm) | 13.11 ± 1.27 | 13.17 ± 1.48 | ns |
| Stone weight (g) | 3.03 ± 0.46 | 3.20 ± 0.44 | ns |
| Pulp yield (%) | 95.00 ± 0.74 | 94.73 ± 0.44 | ns |
| pH | 3.46 ± 0.07 | 3.57 ± 0.09 | ns |
| TSS (°Brix) | 8.67 ± 0.83 | 8.37 ± 0.34 | ns |
| TA (g malic acid $L^{-1}$) | 14.64 ± 3.07 | 12.82 ± 1.84 | ns |
| Maturity index (TSS/TA) | 6.21 ± 0.27 | 6.59 ± 0.51 | ns |

In the ANOVA, ns indicates non-significant. TSS: Total Soluble Solids; TA: Titratable acidity.

The total yield ranged from 29.8 kg $tree^{-1}$ to 31.6 kg $tree^{-1}$ without and with a net, respectively. In this study, the 'Mikado' cultivar grown under a protective net or without a net showed desirable fruit sizes for the consumers, with the fruit weight ranging from 60.93 g (without net) to 61.24 g (with net). The 'Mikado' cv apricot fruits were characterized by a very small stone (<3.2 g) and higher pulp yield (>94.7%). Color plays an essential role in food appearance and acceptability. As shown in Table 2, the external color of the apricot fruits was not affected by cropping systems.

**Table 2.** External color coordinates of 'Mikado' apricot cultivar grown under protective net and without net. Mean values $\pm$ standard deviation ($n = 30$).

| Parameter | 'Mikado' under Protective Net | 'Mikado' without Net | ANOVA |
|:---:|:---:|:---:|:---:|
| L* | 63.37 $\pm$ 1.56 | 63.15 $\pm$ 1.82 | ns |
| a* | 21.60 $\pm$ 2.03 | 22.31 $\pm$ 1.96 | ns |
| b* | 39.92 $\pm$ 1.90 | 39.77 $\pm$ 2.27 | ns |
| C* | 45.75 $\pm$ 1.58 | 46.08 $\pm$ 1.36 | ns |
| h* | 61.30 $\pm$ 3.17 | 60.16 $\pm$3.68 | ns |

In the ANOVA, ns indicates non-significant.

The apricots fruits studied were characterized by red blush on a light orange background skin color, with hue angle values ranging from 60.16 (without net) to 61.30 (with net), L* values from 63.15 (without net) to 63.37 (with net), a* values from 21.60 (with net) to 22.31 (without net). The citric acid, malic acid, fructose, and sucrose contents of apricot fruit were not influenced by the cropping systems; the cropping system effect was significant on the glucose content (Table 3). The main organic acids of apricots 'Mikado' cv were citric (8.7 g kg$^{-1}$ without net-9.3 g kg$^{-1}$ with net) and malic (5.0 g kg$^{-1}$ without net-5.8 g kg$^{-1}$ with net) acids.

**Table 3.** Sugars and organic acids profile, antioxidant activity, and total polyphenols of 'Mikado' apricot grown under protected net and without net. Mean values $\pm$ standard deviation ($n = 30$).

| Parameter | 'Mikado' under Protective Net | 'Mikado' without Net | ANOVA |
|:---:|:---:|:---:|:---:|
| Glucose (g kg$^{-1}$) | 27.2 $\pm$ 0.38 | 25.8 $\pm$ 0.25 | * |
| Fructose (g kg$^{-1}$) | 25.2 $\pm$ 0.19 | 24.2 $\pm$ 0.05 | ns |
| Sucrose (g kg$^{-1}$) | 44.9 $\pm$ 0.14 | 43.7 $\pm$ 0.20 | ns |
| Citric acid (g kg$^{-1}$) | 9.3 $\pm$ 0.11 | 8.7 $\pm$0.08 | ns |
| Malic acid (g kg$^{-1}$) | 5.8 $\pm$ 0.00 | 5.0 $\pm$0.00 | ns |
| ABTS (mmol Trolox kg$^{-1}$ fw) | 2.2 $\pm$ 0.08 | 1.3 $\pm$ 0.01 | * |
| DPPH (mmol Trolox kg$^{-1}$ fw) | 6.2 $\pm$ 0.12 | 4.8 $\pm$ 0.11 | ns |
| FRAP (mmol Trolox kg$^{-1}$ fw) | 6.2 $\pm$ 0.13 | 4.9 $\pm$ 0.11 | ns |
| TP (mg of gallic acid 100 g$^{-1}$ fw) | 16.2 $\pm$ 0.65 | 15.5 $\pm$ 0.83 | ns |

In the ANOVA, ns indicate non-significant and while * significant differences at 0.05.

As for sugars, sucrose was the most abundant sugar in apricots 'Mikado' cv (43.7 g kg$^{-1}$ without net–44.9 g kg$^{-1}$ with net), followed by glucose (25.8 g kg$^{-1}$ without net–27.2 g kg$^{-1}$ with net), and fructose (24.2 g kg$^{-1}$ without net–25.2 g kg$^{-1}$ with net). As can be observed in Table 3, the levels of TP ranged between 15.5–16.2 mg GAE 100 g$^{-1}$ fw (without and with a net, respectively), while the antioxidant activity values determined with three different methods (ABTS, DPPH, and FRAP) ranged between: 1.3 and 2.2 mmol Trolox kg$^{-1}$ fw for ABTS, 4.8 and 6.2 mmol Trolox kg$^{-1}$ fw for DPPH, and 4.9 and 6.2 mmol Trolox kg$^{-1}$ fw for FRAP, without and with a net, respectively. There were no significant differences for all studied parameters related to flavor or texture.Color attributes (measured with colorimeter) were not affected by the use of a net, as shown in Table 2. However, the results of the panel of tasters observe significant differences in terms of external color and color homogeneity. This may be due to the fact that apricots do not have a homogeneous color throughout the fruit, having more orange or reddish parts (this is called 'veneer'). This is perceived by the tasters, while the objective measurement with the colorimeter makes an average of the data taken in the equatorial diameter (Table 4).

**Table 4.** Sensory evaluation. Mean values (*n* = 8).

| Variable | 'Mikado' under Protective Net | 'Mikado' without Net | Significance |
|---|---|---|---|
| Appearance | | | |
| External color | 8.9 | 6.5 | *** |
| Lightness | 4.1 | 5.9 | ** |
| Color homogeneity | 8.5 | 4.3 | *** |
| Internal color | 8.0 | 5.9 | *** |
| Taste, Flavor | | | |
| Sweetness | 3.6 | 3.2 | ns |
| Acidity | 6.6 | 6.5 | ns |
| Astringency | 1.8 | 2.1 | ns |
| Toothech | 1.6 | 1.6 | ns |
| Apricot (flavor) | 3.4 | 2.8 | ns |
| Green (flavor) | 2.8 | 2.4 | ns |
| Ripe fruit (flavor) | 2.2 | 2.0 | ns |
| Vegetable (flavor) | 1.1 | 1.6 | ns |
| Fruit (flavor) | 2.7 | 2.4 | ns |
| Floral (flavor) | 2.1 | 1.9 | ns |
| Texture | | | |
| Skin hardness | 4.0 | 5.1 | ns |
| Hardness | 3.4 | 3.5 | ns |
| Juiciness | 6.1 | 5.4 | ns |
| Pulp consistency | 3.7 | 4.7 | ns |

Trained panelists used a scale from 0 = no intensity to 10 = extremely strong intensity. 'ns' indicates non-significant differences, **, and *** indicate significant differences at *p* < 0.05, 0.01, and 0.001, respectively.

## 4. Discussion

Due to the inexistence of scientific reports concerning apricot cultivation under protective nets, a comparison and discussion of results become difficult. Therefore, the results have been compared with other studies on apricots but grown outdoors. There were no significant differences in the total yield, with a mean of 31.64 kg tree$^{-1}$, for the trees under mesh and of 29.80 kg tree$^{-1}$ in the 'Mikado' trees grown without mesh (Table 1)

The consumers' acceptability, quality, and yield fruit are directly impacted by the fruit weight. In this study, the fruit weight ranged between 60.93 g (without a net) to 61.24 g (with a net), where the values were in agreement with other scientific works focused on apricot [19,20], although these previous studies indicated a high variability of fruit weight among cultivars [19,21,22]. The TSS content is an important quality attribute, as it notably influences the fruit taste. The TSS values of 'Mikado' cv oscillated from 8.37 °Brix (without net) to 8.67 °Brix (with net). Although these results showed lower values than previously reported on apricot by Ruiz and Egea [21] and Mratinić et al. [20], the results could be considered adequate when compared to the bibliography (commonly between 8.6 and 16 °Brix) [4,23,24]. These reported differences may be due to the different environmental conditions, the genotype studied and the precocity of the variety. The flesh firmness varied from 1.72 kg cm$^{-2}$ (without net) to 2.07 kg cm$^{-2}$ (with net), a suitable range for fresh apricot fruits [20]. The titratable acidity (TA) values ranged from 12.82 (without net) to 14.64 (with net) g malic acid L$^{-1}$. Our values of TA are in agreement with Melgarejo et al. [4] for early-apricot "Mirlo anaranjado" cultivar; also, these acidity values were in agreement with Caliskan et al. [25] for early and late apricot varieties. Lo Bianco et al. [24] indicated that early and late-ripening apricot cultivars had the highest acidity contents. Therefore, the maturity index (TSS/TA), an analytical parameter directly associated to the quality of the fruit, in regard to the perception of sweetness and flavor [26], ranged from 6.21 (with net) to 6.59 (without net). Consumer acceptance and both fruit maturity and harvest date are indicated by the fruit color. With respect to the latter, the 'Mikado' cv fruits showed a light orange skin background color (hue value = 60.16, with color space coordinates L* = 63.15,

a* = 22.31, and b* = 39.77 cultivated without a net; fruits while cultivated under a protective net showed a hue value = 61.3, with color space coordinates L* = 63.37, a* = 21.6, and b* = 39.92). These values were an agreement with those obtained by Egea et al. [27] for the varieties "Estrella" and "Sublime" from a medium maturing apricot, but also with early maturing apricot cultivars that showed high L * values [28].

Based on the literature, citric and malic acid are the principal organic acids in apricots [29]. In this context, both were determined for apricot 'Mikado' cv; with citric acid being the predominant organic acid, with values ranging from 8.7 g kg$^{-1}$ (without a net) to 9.3 g kg$^{-1}$ (with a net). The sweet taste is an important quality parameter for fruits, and it is usually associated with sucrose, glucose, and fructose content. In this study, the predominant sugar was sucrose, followed by fructose and glucose. The sucrose, fructose and glucose concentrations averaged between 43.7 (without net) to 44.9 (with net), 24.2 (without net) to 25.2 (with net) and 25.8 (without net) to 27.2 (with net), g kg$^{-1}$, respectively. These results were in agreement with the other studies, suggesting that the major sugars of apricot cultivars were sucrose, glucose, and fructose, respectively [4,25,28]. Although the sugars and acids profile can be affected by the environmental and cultivation factors, together with the genetic factors, the experimental values were considered to be constant [30,31].

Due to the numerous phenolic compounds in apricot, the apricot fruits has important antioxidant, anti-inflammatory, and immunostimulatory activities with positive effects on human health [14]. Regarding the values of the total polyphenols (TP) and total antioxidant activity (TAA) statistically significant differences were observed between the two different cultivation systems. The TAA was higher in 'Mikado' cv cultivated under protective nets. This could be due to climate differences between cultivation under protective nets and outdoors. Under the protective net, the temperature was higher. These results are in accordance with previous papers, which reported that high environmental temperatures could induce the accumulation of the polyphenolic compounds in apricot tree tissues [32].

The TP content of the 'Mikado' cv was lower than that of other studies on commercial apricot cultivars [33], but according to Caliskan et al. [25] and Fan et al. [12], total phenolics of early apricot cultivars are low. The fruit antioxidant capacity and/or the individual quantity of antioxidant compounds were influenced by different factors such as the geographic location of the cultivation region, genotype, and the fruit development time, among others [33,34]. The color is more intense both externally and internally, and also more homogeneous. It has also been observed that the fruits are darker when grown under nets.

## 5. Conclusions

This study was conducted to evaluate the variations in the chemical composition and quality indices of apricots fruits of the 'Mikado' variety as a function of their cultivation under a protective net or outdoors (without nets). This study was a continuation of a practical agronomic assay, once the viability of apricot fruit cultivation under protective nets was confirmed. The results obtained showed that neither the production nor the functional properties of the extra-early variety of apricot 'Mikado' were affected by its cultivation under a protective net. No significant differences were found for all studied parameters related to flavor or texture. However, the cultivation under protective nets allowed minimizing the risk of damage by wind and hail; likewise, the consumption of irrigation water was reduced, soil erosion was reduced, and chemical treatments against pests could be reduced (data not published). Therefore, the technique of protected under net for the cultivation of extra-early apricot 'Mikado' cv is a profitable alternative for farmers, and their use does not affect either the production or the quality.

**Author Contributions:** All the authors contributed in the drafting of the manuscript. P.M., P.L., F.H., R.M.-F., and J.J.M.-N. performed the analysis. P.M., P.L., J.J.M.-N., J.S.S., and Á.A.C.-B. carried out the statistical analysis and interpreted the results. All authors have read and agreed to the published version of the manuscript.

**Funding:** This research received no external funding.

**Conflicts of Interest:** The authors declare no conflict of interest.

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
