# Peer review of "Response of Apricot Fruit Quality to Protective Netting"

_agriculture, doi:10.3390/agriculture11030260_

Round 1

Reviewer 1 Report

General comments

The statistical method needs to be checked. Comments and discussion with sensory evaluation are missing. The comments in the results and discussion section need to be corrected according to statistically significant differences between the two treatments.

Other comments are in attached manuscript.

Author Response

Here are in details the changes made in the manuscript.

Reviewer 2 Report

The article is good and well written.

The little criticism I can make is that iIn some cases, the authors report that there are differences and then say that they are not statistically significant. In these cases, it is worth saying that there are no differences.
I make some suggestions for small corrections, which I send highlighted in the pdf file.

Author Response

Here are in detail the changes made.
